

# Are incremental exercise relationships between rating of perceived exertion and oxygen uptake or heart rate reserve valid during steady-state exercises?

Carlo Ferri Marini[1,*], Lorenzo Micheli[1,*], Tommaso Grossi[1], Ario Federici[1], Giovanni Piccoli[1], Luca Zoffoli[1,2], Luca Correale[3], Stefano Dell'Anna[3,4], Carlo Alberto Naldini[5], Francesco Lucertini[1] and Matteo Vandoni[5]

[1] Department of Biomolecular Sciences –Division of Exercise and Health Sciences, University of Urbino Carlo Bo, Urbino, PU, Italy

[2] Scientific Research & Innovation Department, Technogym S.p.A, Cesena, FC, Italy

[3] Sports Science Unit, Department of Public Health, Experimental and Forensic Medicine, University of Pavia, Pavia, PV, Italy

[4] Department of Theoretical and Applied Sciences, eCampus University, Novedrate, CO, Italy

[5] Laboratory of Adapted Motor Activity (LAMA), Department of Public Health, Experimental and Forensic Medicine, University of Pavia, Pavia, PV, Italy

[*] These authors contributed equally to this work.

Corresponding author
Carlo Ferri Marini,
carlo.ferrimarini@uniurb.it,
c.ferri.marini@umcg.nl

## ABSTRACT

**Background.** Rating of perceived exertion (RPE) is considered a valid method for prescribing prolonged aerobic steady-state exercise (SSE) intensity due to its association with physiological indicators of exercise intensity, such as oxygen uptake ($\dot{V}O_2$) or heart rate (HR). However, these associations between psychological and physiological indicators of exercise intensity were found during graded exercise tests (GXT) but are currently used to prescribe SSE intensity even though the transferability and validity of the relationships found during GXT to SSE were not investigated. The present study aims to verify whether (a) RPE-HR or RPE-$\dot{V}O_2$ relations found during GXTs are valid during SSEs, and (b) the duration and intensity of SSE affect these relations.

**Methods.** Eight healthy and physically active males (age $22.6 \pm 1.2$ years) were enrolled. On the first visit, pre-exercise (during 20 min standing) and maximal (during a GXT) HR and $\dot{V}O_2$ values were measured. Then, on separate days, participants performed 4 SSEs on the treadmill by running at 60% and 80% of the HR reserve (HRR) for 15 and 45 min (random order). Individual linear regressions between GXTs' RPE (dependent variable) and HRR and $\dot{V}O_2$ reserve ($\dot{V}O_2R$) values (computed as the difference between maximal and pre-exercise values) were used to predict the RPE associated with %HRR ($RPE_{HRR}$) and %$\dot{V}O_2R$ ($RPE_{\dot{V}O2R}$) during the SSEs. For each relation (RPE-%HRR and RPE-%$\dot{V}O_2R$), a three-way factorial repeated measures ANOVA ($\alpha = 0.05$) was used to assess if RPE (dependent variable) was affected by exercise *modality* (*i.e.*, RPE recorded during SSE [$RPE_{SSE}$] or GXT-predicted), *duration* (*i.e.*, 15 or 45 min), and *intensity* (*i.e.*, 60% or 80% of HRR).

**Results.** The differences between $RPE_{SSE}$ and GXT-predicted RPE, which were assessed by evaluating the effect of *modality* and its interactions with SSE *intensity* and *duration*, showed no significant differences between $RPE_{SSE}$ and $RPE_{HRR}$. However, when $RPE_{SSE}$ was compared with $RPE_{\dot{V}O2R}$, although *modality* or its interactions with *intensity* were

not significant, there was a significant ($p = 0.020$) interaction effect of *modality* and *duration* yielding a dissociation between changes of $RPE_{SSE}$ and $RPE_{\dot{V}O2R}$ over time. Indeed, $RPE_{SSE}$ did not change significantly ($p = 0.054$) from SSE of 15 min (12.1 ± 2.0) to SSE of 45 min (13.5 ± 2.1), with a mean change of 1.4 ± 1.8, whereas $RPE_{\dot{V}O2R}$ decreased significantly ($p = 0.022$) from SSE of 15 min (13.7 ± 3.2) to SSE of 45 min (12.4 ± 2.8), with a mean change of −1.3 ± 1.5.

**Conclusion**. The transferability of the individual relationships between RPE and physiological parameters found during GXT to SSE should not be assumed as shown by the results of this study. Therefore, future studies modelling how the exercise prescription method used (*e.g.*, RPE, HR, or $\dot{V}O_2$) and SSE characteristics (*e.g.*, exercise intensity, duration, or modality) affect the relationships between RPE and physiological parameters are warranted.

# INTRODUCTION

International organizations (*Brown et al., 2013*; *Haskell et al., 2007*; *O'Donovan et al., 2010*; *US Department of Health, 2018*; *Weggemans et al., 2018*) recommend performing structured and individually tailored aerobic exercise to improve or maintain cardiorespiratory fitness (CRF), which is associated with health status. Aerobic exercise benefits depend on how exercise prescription's variables, represented by the FITT-VP (*i.e.*, frequency, intensity, type, time, volume, and progression) principle, are manipulated (*ACSM, 2021*). In this respect, exercise intensity is pivotal in exercise prescription and its derived benefits. Indeed, to maximize the benefits deriving from an aerobic exercise program (*e.g.*, improvements in CRF) while minimizing the associated risks, it is recommended to prescribe training intensities within certain ranges of minimum and maximal exercise intensities (*ACSM, 2021*). The traditional approach utilized for prescribing aerobic exercise intensity has been to use a percentage of maximal oxygen consumption ($\dot{V}O_{2max}$) or maximal heart rate ($HR_{max}$) (*Mann, Lamberts & Lambert, 2013*). However, previous studies argued that is preferable to prescribe exercise based on reserve values (*i.e.*, the difference between maximal and resting values) to take into account the difference in resting metabolic rate and heart rate (*Swain & Leutholtz, 1997*; *Swain et al., 1998*). Although aerobic exercise can be prescribed based on exercise intensity domains (*i.e.*, moderate, heavy, and severe (*Iannetta et al., 2020*; *Whipp, 1996*)), which are divided by physiological demarcation points (*e.g.*, maximal metabolic steady-state or critical power and lactate or ventilatory threshold) that provide reliable discrimination of the exercise metabolic stimulus (*Iannetta et al., 2020*), the recommended methods proposed by international organizations (*e.g.*, *ACSM, 2021*) for aerobic exercise prescription is still based on fixed percentages of reserve (*i.e.*, heart rate reserve (HRR), oxygen uptake reserve ($\dot{V}O_2R$)) and maximal values (*i.e.*, $HR_{max}$, $\dot{V}O_{2max}$). Although it is still a matter of debate (*Brawner, Keteyian & Ehrman, 2002*; *Ferri Marini et al., 2022a*; *Ferri Marini et al.,*

*2021b*; *Hui & Chan, 2006*; *Pinet et al., 2008*; *Swain et al., 1998*), the exercise prescription modality based on percentages of the reserve values is based on the assumption that, during incremental exercise, %HRR and %$\dot{V}O_2$R have a strong correlation and that their regression line is no different from the identity line (*i.e.,* slope = 1 and intercept = 0) (*Brawner, Keteyian & Ehrman, 2002*; *Byrne & Hills, 2002*; *Colberg, Swain & Vinik, 2003*; *Dalleck & Kravitz, 2006*; *Davenport et al., 2008*; *Lounana et al., 2007*; *Swain & Leutholtz, 1997*). From a practical standpoint, the utilization of this relationship is appealing because it implies that it is possible to elicit a certain metabolic stimulus, namely %$\dot{V}O_2$R, by exercising at the same percentages of HRR. Additionally, the transferability of the relationships between HRR and $\dot{V}O_2$R from incremental to prolonged exercise is another important component that needs to be addressed, which is still subject to debate (*Cunha et al., 2011*; *Wingo, 2015*; *Wingo, Ganio & Cureton, 2012a*). In this respect, previously published articles have demonstrated that the duration (*Ferri Marini et al., 2022a*) and intensity (*Teso, Colosio & Pogliaghi, 2022*) of prolonged exercise affect the relationship between HRR and $\dot{V}O_2$R, showing a dissociation between the two reserves and yield higher %HRR than %$\dot{V}O_2$R in exercise having longer duration and higher intensity.

Over the years, other parameters (*e.g.,* rating of perceived exertion (RPE)) have been proposed to prescribe and monitor aerobic exercise intensity because of their association with physiological markers of exercise intensity (*e.g.,* HR, blood lactate, and $\dot{V}O_2$) (*Borg, 1974*; *Irving et al., 2006*; *Noble et al., 1983*; *Robertson et al., 2004*; *Utter et al., 2004*). Indeed, RPE was found to be associated with physiological parameters such as HR and blood lactate concentration, independently of age, gender, medical history, level of physical activity, and exercise modality (*Scherr et al., 2013*); thus, RPE is considered as a reliable measure to monitor and prescribe aerobic exercise (*Borg, 1998*; *Dunbar et al., 1998*; *Dunbar et al., 1992*; *Eston & Williams, 1988*; *Glass, Knowlton & Becque, 1992*; *Robertson & Noble, 1997*). The American College of Sports Medicine (ACSM) guidelines recommend using two RPE scales: the category scale, also known as the original Borg scale, which rates exercise intensity from 6 to 20, and the category-ratio scale of 0–10, which rates exercise intensity from 0 to 10. In this regard, ACSM guidelines recommend using the 6–20 version scale to prescribe moderate (RPE between 12 and 13) or vigorous (RPE between 14 to 17) aerobic intensities (*ACSM, 2021*). Perceived exertion scales are considered a tool able to consider both physiological and psychological factors during exercise, representing an excellent exercise effort indicator (*Morgan, 1994*). However, the relationships between RPE and HR, $\dot{V}O_2$, or blood lactate (*Borg, Hassmen & Lagerstrom, 1987*; *Borg, 1974*; *Dunbar et al., 1992*; *Glass, Knowlton & Becque, 1992*; *Irving et al., 2006*; *Noble et al., 1983*; *Robertson et al., 2004*; *Utter et al., 2004*) were found during incremental exercises but are used to prescribe and monitor exercise intensity during prolonged constant-intensity exercises. Additionally, there is a high interindividual variability in the relationship between %HRR and %$\dot{V}O_2$R (*Ferri Marini et al., 2023*), which makes the use of standardized values for the entire population potentially inaccurate at an individual level. Indeed, ACSM guidelines (*ACSM, 2012*; *ACSM, 2021*) point out that the RPE responses across exercise modalities and individuals are not comparable due to the high interindividual variability. Therefore, to better individualize the aerobic exercise prescription, international preeminent organizations (*ACSM, 2012*;

*Schoenfeld et al., 2021*) recommend that individuals should be familiarized with the use of RPE during an incremental exercise test and that the RPE corresponding to the desired exercise training intensities should be pointed out during the incremental test. However, during prolonged constant-intensity aerobic exercise, acute physiological adaptations (*i.e.,* cardiovascular drift and oxygen uptake slow component), which lead to increases in HR and $\dot{V}O_2$ values over time and a dissociation between HR and $\dot{V}O_2$ values (*Zuccarelli et al., 2018*), could also alter the RPE relations with HR or $\dot{V}O_2$. Consequently, since RPE-based aerobic exercise prescription relies on relationships between RPE and HR or $\dot{V}O_2$ found during incremental exercises (*ACSM, 2012*; *Schoenfeld et al., 2021*), which have not been verified during prolonged exercises, their use during prolonged exercise lacks external validity and could lead to different metabolic stimuli than predicted. Indeed, using subjective methods for determining exercise intensity contains some pitfalls, and it requires prior maximal or submaximal exercise testing to anchor the RPE associated with the desired training intensity (*e.g.,* $\dot{V}O_2$) and the need to translate incremental exercise test responses to steady-state training workloads (*Bok, Rakovac & Foster, 2022*). Therefore, in order to individualize the training prescription, it is recommended to characterize and understand more deeply the transferability of the relationship between RPE and HRR and $\dot{V}O_2R$.

Finally, although exercise intensity is commonly prescribed using HR (due to the simplicity and low costs of HR monitors), no study assessed how exercise intensity, duration, and their interaction of prolonged aerobic exercise affect the RPE-HR and RPE-$\dot{V}O_2$ relationships when HR is used to prescribe and monitor aerobic exercise intensity. Therefore, to the best of our knowledge, no study investigates whether the RPE-HR and RPE-$\dot{V}O_2$ relationships derived from incremental exercise may be transferable to prolonged constant-intensity exercise; hence, confirming the accuracy of these relationships in prescribing and monitoring prolonged constant-intensity aerobic exercise is warranted. Thus, the results of the present study, which complete those presented in our previous companion article (*Ferri Marini et al., 2022a*), aim to assess if (a) RPE relationships with physiological parameters (*i.e.,* %HRR and %$\dot{V}O_2R$) derived from incremental exercise can be also applied to prolonged exercise and (b) the duration and intensity of prolonged exercise affect these relations.

## MATERIALS & METHODS

Only the procedures and analyses essential and relevant to the aims of the current study have been included in this section. The readers are referred to the companion article (*Ferri Marini et al., 2022a*) of the present article, from which the data used in the present study were collected, for a more comprehensive explanation of the study procedures.

### Participants
Eight healthy and physically active male participants between the ages 18 and 35 years (mean $\pm$ SD: age 22.6 $\pm$ 1.2 years; height 1.83 $\pm$ 0.08 m, body mass 73.5 $\pm$ 9.9 kg, body mass index 21.9 $\pm$ 1.4 kg/m$^2$, body fat percentage 14.8 $\pm$ 4.0) who have received medical clearance for maximal exercises were enrolled in the present study. Participants

were required to have a minimum of three years of treadmill experience. The recruited participants engaged in aerobic exercise training sessions ranging from three to five times per week with at least 4 h dedicated to moderate or higher aerobic training each week (including a minimum of two hours focused on vigorous intensity). Participants who reported the followings were excluded from the study: use of medications that would change the cardiorespiratory responses to exercise (participants did not use any medications during the study), recent orthopedic or musculoskeletal injuries that could affect performance during testing, or smoking habits or use of similar substances.

The study was compliant with the Declaration of Helsinki and was approved by the University of Urbino Human Research Ethics Committee (approval reference number: VN21-10072019). All participants were informed of the potential risks and inconveniences associated with the testing procedures and gave their written informed consent.

## Experimental design

Each participant performed seven testing sessions separated by at least 3 days: the first 3 were preparatory for the next 4 experimental trials (see Fig. 1 for a graphical representation of the experimental design). On the first testing day, pre-exercise and maximal HR and $\dot{V}O_2$ were measured. Then, 2 practice trials were performed to determine the running speed yielding the desired HRR percentages (*i.e.*, 60% and 80% HRR). The 4 experimental trials consisted of running exercises on a treadmill for 15 or 45 min of steady state exercise (SSE) at 60% and 80% of HRR. The order of the practice and experimental trials was randomly assigned to each participant. The testing sessions were performed in a controlled indoor environment (temperature: 19–24 °C; humidity: 40–60%; altitude = 77 m above the sea level; with no fan) at the same time to minimize possible effect of circadian rhythm on HR and $\dot{V}O_2$ values. As explained in detail in the companion article (*Ferri Marini et al., 2022a*), ingestion of liquids or foods was not allowed during the tests. Additionally, participants were asked to avoid changes in their eating habits, vigorous physical activity, and consumption of alcohol and caffeine the day before and the day of testing. The participants were also told to hydrate and drink plenty of fluids the day before and on the day of the trial, drink 0.5 L of water one hour before the session, and fast at least 3 h before the test. The compliance with the above instructions was assessed through a questionnaire specifically created for this study.

## Assessments and data processing

All the exercise tests of this study were performed on the Matrix T7xe treadmill (Johnson Health Tech Italia Spa, Ascoli Piceno, Italy) with 0% slope. In this study, it was not allowed, except for safety reasons, to hold onto the treadmill bars. $\dot{V}O_2$ and HR of the participants were continuously sampled in each testing session. $\dot{V}O_2$, carbon dioxide production, and pulmonary ventilation were monitored and recorded using the portable gas analysis system COSMED K5 (Cosmed, Rome, Italy), which was set up for breath-by-breath data acquisition. Before each test, the system was calibrated using ambient air (21% $O_2$, 0.03 $CO_2$) and a certified gas mixture (16% $O_2$, 5% $CO_2$; Scott Medical Products™, Plumsteadville, USA). The flowmeter turbine was also calibrated using a 3-L syringe

**Figure 1** **Experimental design and timeline of the non-exercise and exercise (treadmill running at 0% grade) assessments.** SSE, steady-state exercise; HR, heart rate; $\dot{V}O_2$, oxygen uptake; GXT, graded exercise test; VT, verification trial; HRR, heart rate reserve; RPE, rating of perceived exertion; †, GXT performed after SSE with no cessation of exercise or warm-up; &, RPE recorded 15 s before the end of each stage; *, RPE collected at 15 min in the 15-minute SSE and at 45 min in the 45-minute SSE.

according to the instructions of the manufacturer. HR was recorded at heart rate intervals using the Polar V800 HR monitor (Polar Electro Oy, Kempele, Finland).

## Anthropometry and body composition

On the first day, the following anthropometric measurements of the participants were taken barefoot and wearing shorts: height (head in the Frankfurt plane), body mass, and body composition (using bioimpedance analysis; BIA 101, Akern-RJL Systems, FI, Italy).

## Pre-exercise HR and $\dot{V}O_2$

HR and $\dot{V}O_2$ were recorded continuously for 20 min with the participant standing. The 20 min were divided into four 5-min bins and the first bin was excluded (*ACSM, 2021*). Then, the bin with the lowest average, for each variable, was considered as pre-exercise HR or $\dot{V}O_2$ (*Ferri Marini et al., 2022a*).

## Maximal exercise test

On the first day, after recording the pre-exercise values of HR and $\dot{V}O_2$, each subject performed 3 min of warm-up at the intensity corresponding to 40% of the maximum estimated speed and immediately performed the control graded exercise test ($GXT_{cont}$). The $GXT_{cont}$ was created using a personalized ramp protocol designed according to the indication proposed by Da Silva and colleagues (*Da Silva et al., 2012*), which consisted in persolized increments in treadmill speed according to participants estiatmed $\dot{V}O_{2max}$ every minute till exhaustion. The personalized GXT was created using the spreadsheet provided by *Ferri Marini et al. (2021a)*, which contains detailed information regarding the steps and formulas used for creating the GXT. Briefly, $\dot{V}O_{2max}$ was estimated using the non-exercise model of *Matthews et al. (1999)*. Then, the initial speed of the incremental exercise protocol was set at 50% of the final speed, and the final speed was calculated from the estimated $\dot{V}O_{2max}$ according to the current ACSM equation (*ACSM, 2021*). The speed was increased every minute by a designated operator. The speed increment value was calculated as the difference between the final and initial speed, which was first divided by 10 min (*i.e.,* the desired test duration), then it was multiplied by the number of minutes passed from the start of the test (excluding warm-up) to the beginning of that stage (*e.g.,* the speeds

increment after 1 min of GXT was computed as: (final speed − initial speed)/10 × 1), please see *Ferri Marini et al. (2021a)* for a detailed explanation of the formulas. This system should allow reaching the final speed and thus the estimated $\dot{V}O_{2max}$, approximately around the tenth minute of testing (*Da Silva et al., 2012*). As explained in detail in the companion article (*Ferri Marini et al., 2022a*), once the GXT$_{cont}$ was completed, participants sat for 20 min (*Nolan, Beaven & Dalleck, 2014*) after which performed a verification trial (VT). The VT consisted in a warm-up of 2 min at 50% of the maximum speed reached during GXT$_{cont}$ followed by 1 min at 70% of the maximum speed reached during GXT$_{cont}$. Finally, the treadmill speed was set at 105% of the maximum speed achieved during GXT$_{cont}$ and maintained till exhaustion. The highest $\dot{V}O_2$ and HR, recorded during GXT$_{cont}$ or VT, were considered maximum values if a $\dot{V}O_2$ plateau was present or if the highest HR recorded during GXT$_{cont}$ and VT were within 4 bpm (*Midgley & Carroll, 2009*). If the data obtained did not meet at least one of these criteria, the test was repeated. During GXT$_{cont}$ and VT, each participant received strong verbal encouragement to make his maximum effort. During GXT$_{cont}$, the 6–20 RPE values were taken at each stage, 15 s before moving on to the next stage.

## Practice trials

Two trials were performed at 60% and 80% of HRR to verify whether the speeds established with the ACSM running equation (*ACSM, 2021*) would elicit the actual HRR values desired. The practice trials started with 3 min of warm-up at speed corresponding to 40% of $\dot{V}O_2R$; then, the intensity was increased to a speed corresponding to either 60% or 80% of $\dot{V}O_2R$ (random order), which, given the 1:1 relationship between %HRR and %$\dot{V}O_2R$, should correspond to 60% and 80% HRR. After 3 min, necessary to reach HRs close to the desired intensity (*i.e.,* 60% and 80% HRR), the speed was adjusted (when needed) every 30s to find the speed yielding the desired target %HRR, which was then used as starting exercise intensity of the SSEs. The practice trials allowed a more accurate determination of the speed yielding the desired HR, which could have been biased by the lack of the 1:1 relationship between %HRR and %$\dot{V}O_2R$ (*Ferri Marini et al., 2022a*; *Ferri Marini et al., 2021b*; *Ferri Marini et al., 2023*) or the presence of errors in the ACSM's running equation. The practice trials were intended to last less than 12 min to avoid possible altered HR responses due to the onset of cardiovascular drift (*Wingo, 2015*).

## Steady-state exercise

After performing the practice trials, each participant performed four SSE trials at different intensities and speeds. All SSEs began with 5 min of warm-up at a speed corresponding to 40% $\dot{V}O_2R$, calculated using the ACSM running equation (*ACSM, 2021*), followed by 15 or 45 min of run in SSE at 60% or 80% HRR. After the warm-up, the intensity gradually increased every 30 s to reach the starting speed found in the practice trials in 150 s. After about 2 min of running at the starting speed, treadmill belt speed was adjusted to maintain the target HR throughout the SSE session. At the end of the experimental trial, participants immediately (without pause) performed the GXT (GXT$_{post}$), already performed on the first day (without the 3-minute warm-up phase) to measure $\dot{V}O_{2peak}$. The maximal reached
values of HR and $\dot{V}O_2$ were used to compute the SSEs' %HRR-$\dot{V}O_2$R relationships (*Wingo, Ganio & Cureton, 2012a*). As explained in *Ferri Marini et al. (2022a)* the testing session was repeated if a steady exercise intensity was not found during the SSE (*i.e.*, the RMSE between the actual and target HR of the trial was higher of 4 bpm) or if either a $\dot{V}O_2$ plateau was not identified during the GXT$_{post}$ (*Midgley & Carroll, 2009*) or the highest HR was not within 4 bpm from the HR assumed to be maximal in the pre-SSE session (a total of 3 experimental trials were repeated). During SSE, RPE data (RPE$_{SSE}$) were collected at 15 min in the 15-minute SSE and at 45 min in the 45-minute SSE, using 6–20 scale (*Borg, 1998*). RPE$_{SSE}$ is the actual value of perceived effort at the end of the SSE.

## Data preparation and processing

Before the analyses were carried out, dataset preparation, procedure, and data processing were performed. The stationary averages of the last 30-sec of the $\dot{V}O_2$ and HR values recorded for each stage of the GXT$_{cont}$ were computed as a percentage of the reserve values using the following formula: $100 \times$ (recorded value − pre-exercise value) / (maximal value − pre-exercise value). The averages of the last 5 min of the 15- and 45-minute trials were considered as steady-state values of HR and $\dot{V}O_2$, which were converted to percentage values of HRR and $\dot{V}O_2$R (SSE %HRR and SSE %$\dot{V}O_2$R) using pre-exercise data and peaks recorded during the GXT$_{post}$ following the corresponding SSE. RPE-%HRR and RPE-%$\dot{V}O_2$R relationships obtained during GXT$_{cont}$ were used to estimate RPE values during SSE at 60%HRR and 80%HRR, based on SSE %HRR (RPE$_{HRR}$) and %$\dot{V}O_2$R (RPE$_{\dot{V}O2R}$) as follows. First, RPE and %$\dot{V}O_2$R or %HRR paired data points recorded during the GXT$_{cont}$ (warm-up excluded) were used to perform individual linear regressions between RPE and %$\dot{V}O_2$R and RPE and %HRR for each participant using RPE as dependent variable. The regression coefficients (*i.e.*, intercept and slope) and the goodness of fit of the individual linear regressions are shown in Table 1.

Then, participants' RPEs were predicted using the slopes and the intercepts of the individual linear regression equations between RPE and %HRR or %$\dot{V}O_2$R derived from the GXT$_{cont}$ using the following formulas: slope $\times$ SSE %HRR + intercept; slope $\times$ SSE %$\dot{V}O_2$R + intercept.

## Statistical analysis

The following analyses were used to assess if the individual relationships between RPE-%HRR and RPE-%$\dot{V}O_2$R derived from incremental exercises are different from the relationships found during SSEs of different durations and intensities. Therefore, the RPE predicted from the GXT$_{cont}$ (*i.e.*, RPE$_{\dot{V}O2R}$ and RPE$_{HRR}$) were separately compared to the RPE recorded during different SSEs (*i.e.*, RPE$_{SSE}$) using two three-way factorial repeated measures ANOVAs. The ANOVAs were performed to assess if the RPE (dependent variable) was affected by the following independent variables: (a) the modality used to determine the RPE (*i.e.*, RPE$_{SSE}$ or RPE predicted—*modality*), (b) the intensity of the SSE (*i.e.*, 60 or 80% of HRR—*intensity*), and (c) the duration of the SSE (*i.e.*, 15 or 45 min—*duration*). When a statistically significant result was found, the Bonferroni method was used to perform the post hoc pairwise comparisons. In the present study, three-way factorial repeated measures

**Table 1  The regression coefficients and the goodness of fit of the individual linear regressions (ILR).**

|  | R | R² | SEE | Intercept | Slope |
|---|---|---|---|---|---|
| RPE *vs.* $\dot{V}O_2R$ |  |  |  |  |  |
| Mean | 0.944 | 0.893 | 1.122 | −1.370 | 0.209 |
| SD | 0.047 | 0.087 | 0.484 | 4.939 | 0.051 |
| Min | 0.854 | 0.729 | 0.537 | −10.333 | 0.139 |
| Max | 0.988 | 0.976 | 1.872 | 4.599 | 0.286 |
| RPE *vs.* HRR |  |  |  |  |  |
| Mean | 0.965 | 0.930 | 0.929 | −0.482 | 0.194 |
| SD | 0.021 | 0.040 | 0.297 | 3.871 | 0.038 |
| Min | 0.933 | 0.871 | 0.483 | −8.310 | 0.146 |
| Max | 0.993 | 0.986 | 1.311 | 4.603 | 0.260 |

**Notes.**

R, coefficient of correlation; R², coefficient of determination; SEE, standard error of estimate; Intercept, intercept of the ILR created during control graded exercise test ($GXT_{cont}$); Slope, slope of the ILR created during $GXT_{cont}$; RPE *vs.* $\dot{V}O_2R$, ILR between rating of perceived exertion (RPE) and $\dot{V}O_2R$ recorded during $GXT_{cont}$; RPE *vs.* HRR, ILR between RPE and HRR recorded during $GXT_{cont}$; SD, standard deviation.

ANOVAs were used because this comprehensive approach allows a direct assessment of the interaction effect of exercise intensity and duration, which would be neglected if separate two-way ANOVAs for each exercise intensity (assessing *modality × duration*) or duration (assessing *modality × intensity*) were performed.

Moreover, for each of the four experimental conditions, the Cohen's *d* effect size (ES) between $RPE_{SSE}$ and RPE predicted was calculated by dividing the mean by the SD of the differences between $RPE_{SSE}$ and RPE predicted. Finally, the number of participants whose $RPE_{SSE}$ was not within ± 1 of the predicted RPE were computed to assess, on an individual level, the number of overestimation (*i.e.,* difference between $RPE_{SSE}$ and predicted RPE lower than -1) and underestimation (*i.e.,* difference between $RPE_{SSE}$ and predicted RPE is higher than 1) of the $RPE_{SSE}$. The analyses were performed using SPSS Statistics (IBM, v.20) software, with an $\alpha$ level of 0.05.

# RESULTS

Participants' pre-exercise and maximal HR and $\dot{V}O_2$ values recorded during control tests are: pre-exercise HR = 79.6 ± 10.5 bpm, $HR_{max}$ = 195.3 ± 11.3 bpm, pre-exercise $\dot{V}O_2$ = 4.6 ± 0.5 mL $\cdot min^{-1} \cdot kg^{-1}$, $\dot{V}O_{2max}$ = 61.5 ± 8.6 mL$\cdot min^{-1} \cdot kg^{-1}$.

HR and $\dot{V}O_2$ recoded during the SSEs, along with the actual RPE and the RPE predicted according to the $GXT_{cont}$'s individual linear relationships between RPE and %$\dot{V}O_2R$ or %HRR, are reported in Table 2.

The Mean, SD, and ES values of the differences between the RPE values measured during SSE ($RPE_{SSE}$) and the RPEs predicted using the relationship between RPE and $\dot{V}O_2R$ ($RPE_{\dot{V}O2R}$) or HRR ($RPE_{HRR}$) found during $GXT_{cont}$ are shown in Table 3.

The number of participants whose $RPE_{SSE}$ was not within ± 1 of the predicted RPE, showing, on an individual level, an overestimation or underestimation of the $RPE_{SSE}$ are reported in Table 4.
**Table 2** HR, V̇O₂, and RPE responses to 15 and 45 min of SSE with HR held constant at 60% and 80% of HRR (mean ± SD).

| | 15 min | | 45 min | |
| --- | --- | --- | --- | --- |
| | SSE at 60% of HRR | SSE at 80% of HRR | SSE at 60% of HRR | SSE at 80% of HRR |
| HR (bpm) | $150.8 \pm 9.7$ | $173.4 \pm 10.2$ | $150.1 \pm 9.8$ | $173.0 \pm 10.5$ |
| V̇O₂ (mL · min⁻¹ · kg⁻¹) | $39.5 \pm 6.3$ | $52.5 \pm 5.5$ | $36.4 \pm 6.6$ | $45.1 \pm 4.5$ |
| $HR_{peak}$ (bpm) | $195.2 \pm 12.9$ | $193.3 \pm 10.6$ | $190.9 \pm 12.0$ | $194.5 \pm 10.4$ |
| V̇O₂$_{peak}$ (mL · min⁻¹ · kg⁻¹) | $61.7 \pm 6.6$ | $63.0 \pm 6.8$ | $58.9 \pm 9.4$ | $60.7 \pm 8.7$ |
| $RPE_{SSE}$ | $10.6 \pm 1.2$ | $13.5 \pm 1.7$ | $12.3 \pm 1.7$ | $14.8 \pm 1.7$ |
| $RPE_{V̇O2R}$ | $11.5 \pm 2.7$ | $15.8 \pm 2.1$ | $11.0 \pm 3.0$ | $13.9 \pm 1.8$ |
| $RPE_{HRR}$ | $11.5 \pm 2.1$ | $15.5 \pm 1.5$ | $11.8 \pm 2.1$ | $15.3 \pm 1.5$ |
| $HRR_{peak}$ (%) | $61.7 \pm 2.8$ | $82.6 \pm 1.6$ | $63.4 \pm 1.5$ | $81.4 \pm 2.3$ |
| V̇O₂$R_{peak}$ (%) | $60.9 \pm 6.7$ | $82.1 \pm 5.7$ | $58.6 \pm 7.8$ | $72.7 \pm 6.4$ |

**Notes.**
HR, heart rate; V̇O₂, oxygen uptake; RPE, rating of perceived exertion (6–20); SSE, steady-state exercise; HRR, heart rate reserve; V̇O₂R, oxygen uptake reserve; SD, standard deviation; $HR_{peak}$, peak HR recorded during incremental exercises following each SSE; V̇O₂$_{peak}$, peak V̇O₂ recorded during incremental exercises following each SSE; $RPE_{SSE}$, rating of perceive exertion recorded during SSE; $RPE_{V̇O2R}$, rating of perceive exertion predicted from SSE V̇O₂R using incremental RPE-V̇O₂R relationship; $RPE_{HRR}$, rating of perceive exertion predicted from SSE HRR using incremental RPE-HRR relationship.

**Table 3** Mean, SD, and ES of the differences between actual and predicted RPE values.

| | $RPE_{SSE} - RPE_{V̇O2R}$ | | $RPE_{SSE} - RPE_{HRR}$ | |
| --- | --- | --- | --- | --- |
| | 15 min | 45 min | 15 min | 45 min |
| SSE at 60% of HRR | | | | |
| Mean | $-0.9$ | $1.3$ | $-0.9$ | $0.5$ |
| SD | $2.8$ | $3.2$ | $2.1$ | $2.3$ |
| ES | $-0.32$ | $0.40$ | $-0.42$ | $0.21$ |
| SSE at 80% of HRR | | | | |
| Mean | $-2.3$ | $0.9$ | $-2.0$ | $-0.5$ |
| SD | $3.2$ | $2.7$ | $2.2$ | $2.2$ |
| ES | $-0.74$ | $0.34$ | $-0.93$ | $-0.23$ |

**Notes.**
SD, standard deviation; ES, Cohen's $d$ effect size; HRR, heart rate reserve; V̇O₂R, oxygen uptake reserve; SSE, steady-state exercise; $RPE_{SSE}$, rating of perceived exertion (RPE) reported during SSE; $RPE_{V̇O2R}$, rating of perceive exertion predicted from SSE V̇O₂R using incremental RPE-V̇O₂R relationship; $RPE_{HRR}$, rating of perceive exertion predicted from SSE HRR using incremental RPE-HRR relationship.

## $RPE_{SSE}$ *vs* $RPE_{V̇O2R}$

Measured and V̇O₂R estimated RPEs were affected by *intensity* ($F_{(1,7)} = 68.744$, $p < 0.001$, partial eta-squared ($\eta_p^2$) $= 0.908$), but they were not affected by the *modality* ($F_{(1,7)} = 0.088$, $p = 0.775$, $\eta_p^2 = 0.012$) and *duration* ($F_{(1,7)} = 0.081$, $p = 0.784$, $\eta_p^2 = 0.011$).

The interactions between *duration × modality* ($F_{(1,7)} = 9.030$, $p = 0.020$, $\eta_p^2 = 0.563$) and *duration × intensity* ($F_{(1,7)} = 6.676$, $p < 0.036$, $\eta_p^2 = 0.488$) showed a significant effect on RPE, whereas the interactions between *intensity × modality* ($F_{(1,7)} = 2.394$, $p = 0.166$, $\eta_p^2 = 0.255$) and *intensity ×duration ×modality* ($F_{(1,7)} = 4.778$, $p = 0.065$, $\eta_p^2 = 0.406$) were not significant.

RPE was higher at higher intensity, with a significant mean difference between 80% of HRR ($14.5 \pm 2.0$) and 60% of HRR ($11.3 \pm 2.2$) of $3.1 \pm 1.6$, $p < 0.001$.

**Table 4** Number of participants (N) whose differences between actual and predicted RPE values ($\Delta$) was not within ±1.

| | RPE$_{SSE}$ - RPE$_{\dot{V}O2R}$ | | RPE$_{SSE}$ - RPE$_{HRR}$ | |
| --- | --- | --- | --- | --- |
| | 15 min | 45 min | 15 min | 45 min |
| SSE at 60% of HRR | | | | |
| $\Delta$ <-1 (N) | 3 | 2 | 4 | 3 |
| $\Delta$ >1 (N) | 2 | 3 | 1 | 3 |
| SSE at 80% of HRR | | | | |
| $\Delta$ <-1 (N) | 5 | 2 | 5 | 2 |
| $\Delta$ >1 (N) | 1 | 3 | 0 | 2 |

**Notes.**

HRR, heart rate reserve; $\dot{V}O_2R$, oxygen uptake reserve; SSE, steady-state exercise; RPE$_{SSE}$, rating of perceived exertion (RPE) reported during SSE; RPE$_{\dot{V}O2R}$, rating of perceive exertion predicted from SSE $\dot{V}O_2R$ using incremental RPE-$\dot{V}O_2R$ relationship; RPE$_{HRR}$, rating of perceive exertion predicted from SSE HRR using incremental RPE-HRR relationship.

When the *modalities* were compared at different SSE durations, RPE$_{SSE}$ did not increase significantly from SSE of 15 min (12.1 ± 2.0) to SSE of 45 min (13.5 ± 2.1), with a mean change of 1.4 ± 1.8, $p = 0.054$. Whereas RPE$_{\dot{V}O2R}$ decreased significantly from SSE of 15 min (13.7 ± 3.2) to SSE of 45 min (12.4 ± 2.8), with a mean change of −1.3 ± 1.5, $p = 0.022$.

### RPE$_{SSE}$ *vs* RPE$_{HRR}$

Measured and HRR-estimated RPEs were only affected by *intensity* ($F_{(1,7)} = 79.933$, $p < 0.001$, $\eta_p^2 = 0.919$), whereas *duration* ($F_{(1,7)} = 4.744$, $p = 0.066$, $\eta_p^2 = 0.404$), *modality* ($F_{(1,7)} = 1.165$, $p = 0.316$, $\eta_p^2 = 0.143$), and the interaction between *duration* × *intensity* ($F_{(1,7)} = 3.117$, $p = 0.121$, $\eta_p^2 = 0.308$), *duration* × *modality* ($F_{(1,7)} = 5.506$, $p = 0.051$, $\eta_p^2 = 0.440$), *modality* × *intensity* ($F_{(1,7)} = 5.089$, $p = 0.059$, $\eta_p^2 = 0.421$), and *duration* × *modality* × *intensity* ($F_{(1,7)} = 0.256$, $p = 0.628$, $\eta_p^2 = 0.035$) did not show a significant effect on RPE.

## DISCUSSION

The results of the present study are the first to assess the transferability of the RPE-%HRR and RPE-%$\dot{V}O_2R$ relationships from incremental to prolonged exercise prescribed using HR, which is one of the underlying assumptions based upon which world's preeminent organizations (*ACSM, 2012*; *Schoenfeld et al., 2021*) recommend using the RPE-based methods for prescribing aerobic exercise intensity. However, it is worth notice that the results of the present study were collected in a small sample of physically active males, which, as explained in detail at the end of the discussion, limits and reduce the generalizability of the findings. The present study demonstrates that, as expected, exercise intensity affects RPE, with higher exercise intensities yielding higher RPEs during both GXTs and SSEs. This result confirms the presence of a correlation between perceived exertion and exercise intensity, which is the underlying theoretical construct that allows the use of RPE scales for prescribing and monitoring exercise intensity during aerobic (*Borg, 1998*; *Dunbar et al., 1998*; *Dunbar et al., 1992*; *Eston & Williams, 1988*; *Glass, Knowlton & Becque, 1992*; *Robertson & Noble, 1997*) and resistance (*Ferri Marini et al., 2022b*; *Gearhart Jr et al., 2002*; *Lagally et al., 2004*;

*Lea et al., 2022; Morishita et al., 2018; Pincivero, Coelho & Campy, 2003*) exercise. A novel result that can be observed in the present study derives from the comparison of $RPE_{\dot{V}O2R}$ and $RPE_{SSE}$, which shows a significant effect of the interaction between duration and prediction modality used on RPE, yielding a dissociation, over time, between the actual and the $\dot{V}O_2$ predicted RPE. Indeed, from 15 to 45 min SSE, it is noticeable an increase of $1.4 \pm 1.8$ in $RPE_{SSE}$ and a decrease of $1.3 \pm 1.5$ in $RPE_{\dot{V}O2R}$, showing an opposite trend with increasing duration. Likewise, the interaction effect between prediction modality based on HRR and duration, even though not reaching a statistical significance, showed a tendency towards a significant effect ($p = 0.051$), yielding an increase of $1.4 \pm 1.8$ in $RPE_{SSE}$ with no changes in $RPE_{HRR}$ ($0.0 \pm 0.5$).

Additionally, as shown in Table 3, it is possible to notice a tendency showing that intensity may also affect the relationship between $RPE_{SSE}$ and $RPE_{\dot{V}O2R}$ or $RPE_{HRR}$, with higher intensity SSEs (*i.e.,* 80%HRR) showing mean differences (*i.e.,* $RPE_{SSE}$ minus $RPE_{\dot{V}O2R}$ or $RPE_{HRR}$) from $-0.4$ to $-1.4$ lower than lower intensity SSE (*i.e.,* 60%HRR) for both $RPE_{\dot{V}O2R}$ and $RPE_{HRR}$ in both 15- and 45-min conditions. Since the mean differences between $RPE_{SSE}$ and $RPE_{\dot{V}O2R}$ or $RPE_{HRR}$ tended to be lower at higher intensities, higher SSE intensity caused the incremental exercise relationships between RPE and $\dot{V}O_2R$ or HRR to overestimate the actual $RPE_{SSE}$ compared to the lower SSE intensity.

The results of the present study (Table 3) point out that also the duration may affect the relationship between $RPE_{SSE}$ and $RPE_{\dot{V}O2R}$ or $RPE_{HRR}$ with longer duration SSEs (*i.e.,* 45 min) showing mean differences (*i.e.,* $RPE_{SSE}$ minus $RPE_{\dot{V}O2R}$ or $RPE_{HRR}$) from 1.4 to 3.2 higher than shorter duration SSE (*i.e.,* 15 min) for both $RPE_{\dot{V}O2R}$ and $RPE_{HRR}$ in both 60% and 80% conditions. Since the mean differences between $RPE_{SSE}$ and $RPE_{\dot{V}O2R}$ or $RPE_{HRR}$ tended to be higher at longer durations, longer SSE duration caused the incremental exercise relationships between RPE and $\dot{V}O_2R$ or HRR to underestimate the actual $RPE_{SSE}$ compared to the shorter SSE duration.

When the results of the present study are interpreted on an individual level, assessing the number of participants whose $RPE_{SSE}$ was not within $\pm 1$ of the predicted RPE, the effect of SSE duration and intensity seem to be confirmed, showing that the $RPE_{SSE}$ was overestimated in more participants, with fewer underestimations, in shorter duration and higher intensity SSEs.

Overall, the above results highlight that when aerobic exercise intensity is prescribed using a fixed %HRR, predicting RPE using RPE-%$\dot{V}O_2R$ relationships found during incremental exercise yields inaccurate estimations of the SSE intensity in healthy males, whereas the $RPE_{HRR}$ values do not significantly differ from $RPE_{SSE}$ values. However, it is worth noting that both relationships (*i.e.,* RPE-%HRR and RPE-%$\dot{V}O_2R$) seem to be affected by the duration and intensity of prolonged aerobic exercise.

The results of the present study are in line with those found by previous studies aiming at investigating the suitability of RPE as an aerobic exercise intensity prescription method, which found an association between RPE and physiological parameters during incremental exercise (*Dunbar et al., 1994; Dunbar et al., 1992; Glass, Knowlton & Becque, 1992*). Although similar, the results of the present study partly contrast with those found by Glass and colleagues (*Glass, Knowlton & Becque, 1992*), who tried to assess if RPE derived

from GXT could be used to prescribe SSE intensity based on the RPE-HR relationship and demonstrated that RPE obtained from GXT can accurately prescribe exercise intensity during treadmill running. However, the results of the present study and those from *Glass, Knowlton & Becque (1992)* are not entirely comparable because of the different durations of the exercises (*i.e.,* 15 and 45 min compared to 10 min used from Glass et al.). In particular, the current study highlights that the relationship found during incremental exercise, when used to prescribe prolonged exercise, could be affected by SSE duration. Therefore, the shorter duration (10 min of exercise) used by *Glass, Knowlton & Becque (1992)*, may not be sufficient to see the physiological adjustment in HR (*i.e.,* cardiovascular drift) that occurs approximately after 10–15 min of moderate-intensity exercise (*Wingo, Ganio & Cureton, 2012a*).

The results of the present study show that not all the physiological parameters can be used to predict the actual RPE during SSE in healthy males. Specifically, while $RPE_{SSE}$ and $RPE_{HRR}$ did not change significantly with longer duration, $RPE_{\dot{V}O_2R}$ decreased significantly from shorter to longer SSEs. These results may be due to physiological adjustments present during prolonged exercise, namely the so-called slow components (*Jones et al., 2011*), which affect both $\dot{V}O_2$ and HR (*i.e.,* cardiovascular drift) influencing the transferability of the relationship between HR and $\dot{V}O_2$ from incremental to prolonged exercise and creating a dissociation between the two parameters during prolonged exercise (*Ferri Marini et al., 2022a*; *Teso, Colosio & Pogliaghi, 2022*; *Zuccarelli et al., 2018*). The results of the present study reinforce all the above and, also, reveal discrepancies between RPE and physiological parameters (*i.e.,* $\dot{V}O_2R$ and HRR) when comparing the relationships found during incremental and prolonged exercise, showing that $RPE_{SSE}$ is, on average, closer (*i.e.,* smaller differences with predicted RPEs) to $RPE_{HRR}$ compared to $RPE_{\dot{V}O_2R}$.

A novelty of the present study is that, although there are several studies assessing the use of different indicators of exercise intensity (*e.g.*, HR or $\dot{V}O_2$) during prolonged aerobic exercise (see *Ferri Marini et al., 2022a*), only one (*Wingo & Cureton, 2006b*) employed prolonged aerobic exercise bouts performed at a constant HR, which is a commonly adopted and recommended practice in aerobic exercise prescription. However, *Wingo & Cureton (2006b)* did not assess the compliance to the target SSE intensities, whereas in the present study participants' compliance with the target intensity was assessed by calculating the RMSE between actual and target HR for each subject and SSE condition. Additionally, exercising at constant power output or speed for a prolonged duration yield higher exercise intensity (*i.e.,* HR and $\dot{V}O_2$) due to physiological adjustments like cardiovascular drift and the $\dot{V}O_2$ slow component (*Cunha et al., 2011*; *Teso, Colosio & Pogliaghi, 2022*) and lower $\dot{V}O_{2max}$ (*Wingo, Ganio & Cureton, 2012a*), whereas $HR_{max}$ has been shown not to be modified by prolonged aerobic exercise (*Ganio et al., 2006*; *Lafrenz et al., 2008*; *Wingo & Cureton, 2006a*; *Wingo & Cureton, 2006b*; *Wingo et al., 2012b*; *Wingo, Stone & Ng, 2020*). Therefore, maintaining a constant HR throughout each condition ensures consistent relative exercise intensity across all SSEs unlike using fixed percentages of $\dot{V}O_{2max}$ or $\dot{V}O_2R$ or speeds corresponding to certain $\dot{V}O_2$. During exercise at a fixed %HRR, as proposed in the present study, the discrepancy between $RPE_{SSE}$ and $RPE_{\dot{V}O_2R}$ may also appear because the work rate (*e.g.*, power output in a cycle ergometer or speed and grade in a

treadmill) needs to be reduced as time goes by (*Ferri Marini et al., 2022a*; *Iannetta et al., 2020*) to maintain the target HR and avoid the effects of cardiovascular drift yielding lower $\dot{V}O_2$. Indeed, in the current study, as explained in detail in the companion article (*Ferri Marini et al., 2022a*), treadmill speeds were continuously reduced after reaching target HR to maintain a stable HR during each SSE. The treadmill speeds were decreased of about the same magnitude during the 15-min SSEs at 60% HRR (~4.3%) and 80% HRR (~5.7%), whereas a more noticeable decrease in treadmill speeds was observed during the 45-min SSEs at 80% HRR (~24.1%) compared to SSEs at 60% HRR (~13.2%). The seemingly higher decrease in treadmill speeds in longer duration and higher intensity SSEs suggests a potential interaction effect between exercise intensity and duration on the cardiovascular drift. Consequently, the one-size-fits-all principle (*i.e.,* the principle of applying a standard method for prescribing exercise without considering interindividual variability in response to prolonged exercise) should be carefully adopted as the aerobic exercise intensity seems to be affected not only by the exercise duration but also by the parameter used to prescribe exercise intensity (*e.g.,* RPE, HR, power output, or $\dot{V}O_2$) in healthy male individuals. Indeed, although in the current study individual RPE-%HRR and RPE-%$\dot{V}O_2$R relationships derived from incremental tests were used, this was not sufficient to accurately predict relationships during prolonged exercise in physically active males. Thus, additional methods able to consider metabolic responses, such as metabolic thresholds (*e.g.,* critical power, maximal metabolic steady state), could increase exercise prescription accuracy (*Weatherwax et al., 2019*), and further studies investigating whether different methods of prescribing exercise could affect adaptations to exercise are required. Moreover, considering that the RPE predicted by $\dot{V}O_2$R is lower than the actual RPE found during longer SSE, whereas is higher during shorter SSE, using the standardized relationship to transfer the results found in incremental exercise to prolonged exercise could induce individuals to not exercise at the desired intensity. Therefore, an inaccurate prediction of RPE that would lead to a lower intensity than expected may not lead to the desired adaptations from training. On the contrary, prescribing higher intensity than expected could potentially increase the risk associated with exercise due to the higher cardiorespiratory, metabolic, and mechanical stress at higher exercise intensities.

From a psychological perspective, RPE is also related to dispositional and situational factors (*Morgan, 1973*; *Morgan, 1994*; *Noble & Robertson, 1996*). For this reason, it is necessary to understand how RPE correlates with exercise intensity, especially with high intensity. In this context, a more robust understanding of connections and links between physiological (*e.g.,* HR, $\dot{V}O_2$) and psychological (*i.e.,* RPE) parameters, and the transferability of this relationship during steady-state exercise could also lead to increased accuracy of aerobic exercise intensity prescription. Furthermore, *Hall, Ekkekakis & Petruzzello (2005)* found that the relationship between the factors above becomes weaker as the intensity increases, probably because high exercise intensities are those in which personality and cognitive influences begin to subside.

It is worth noting that the results of the present study were collected in a small and homogenous sample of young, physically active, and healthy males; hence, future studies should assess those relationships in different populations (*e.g.,* females, sedentary or

pathological individuals). The lack of a priori sample size calculation and power analysis is due to the fact that the present study is a retrospective analysis using the data collected in the companion article by *Ferri Marini et al. (2022a)*. However, the results reported in the present study (*i.e.,* effect size (Table 3), over- and under- estimation on an individual level (Table 4), and supplementary individual raw data) can provide preliminary information that may help researchers and practitioners to delve deeper on the topic of the transferability of the relations between physiological and psychological exercise intensity indices from incremental to prolonged exercises.

Finally, a limitation of the use of RPE, which is also present in the present study, is that perceived exertion could be affected by the exercise sessions previously performed. However, in the present study, the presence of a systematic bias caused by this issue should be mitigated by the randomization of the experimental trials and the compliance with the standardized anchoring procedures and pre-test instructions used before each experimental trial, which specifically inform the subject to express their perceived exertion without considering the previously reported values (*ACSM, 2012*; *Schoenfeld et al., 2021*).

In addition, the results of the present study also suggest that it would be advisable not to rely on only one parameter (*i.e.,* HR, $\dot{V}O_2$, power output, or RPE) to prescribe and monitor aerobic exercise intensity but rather to use a combination of multiple parameters. This approach could help researchers and practitioners to increase the aerobic exercise intensity prescription accuracy and the benefit-to-risk ratio of aerobic training.

## CONCLUSIONS

This study adds information to its companion article, showing that prolonged exercise duration and intensity could affect not only the relationship between HRR and $\dot{V}O_2R$, but also their relationship with RPE. Indeed, the transferability of the individual relationships between RPE and physiological parameters (*i.e.,* HRR and $\dot{V}O_2R$) found during incremental exercise to prolonged exercise, hence their validity in prescribing aerobic exercise intensity during prolonged aerobic exercise, was not confirmed in the present study. Therefore, to accurately prescribe and monitor aerobic exercise intensity, exercise professionals should be aware of the physiological and psychological adjustments happening during prolonged exercise, and future studies should assess how these adjustments are affected by the parameters (*i.e.,* HR, $\dot{V}O_2$, power output, or RPE) used to prescribe and monitor exercise intensity. Finally, although RPE represents a valuable method to prescribe exercise intensity, it is recommended that the subjects are familiarized with using it and that, for exercise prescription purposes, RPE is used alongside other parameters.

### Funding
The authors received no funding for this work.

## Competing Interests

Carlo Ferri Marini, Matteo Vandoni, and Luca Correale are Academic Editors for PeerJ. Luca Zoffoli is employed by Technogym S.p.A., Cesena, FC, Italy.

## Author Contributions

- Carlo Ferri Marini conceived and designed the experiments, performed the experiments, analyzed the data, prepared figures and/or tables, authored or reviewed drafts of the article, and approved the final draft.
- Lorenzo Micheli performed the experiments, analyzed the data, prepared figures and/or tables, authored or reviewed drafts of the article, and approved the final draft.
- Tommaso Grossi performed the experiments, prepared figures and/or tables, and approved the final draft.
- Ario Federici conceived and designed the experiments, authored or reviewed drafts of the article, and approved the final draft.
- Giovanni Piccoli conceived and designed the experiments, authored or reviewed drafts of the article, and approved the final draft.
- Luca Zoffoli analyzed the data, authored or reviewed drafts of the article, and approved the final draft.
- Luca Correale performed the experiments, analyzed the data, authored or reviewed drafts of the article, and approved the final draft.
- Stefano Dell'Anna performed the experiments, analyzed the data, prepared figures and/or tables, and approved the final draft.
- Carlo Alberto Naldini performed the experiments, prepared figures and/or tables, and approved the final draft.
- Francesco Lucertini conceived and designed the experiments, prepared figures and/or tables, authored or reviewed drafts of the article, and approved the final draft.
- Matteo Vandoni conceived and designed the experiments, prepared figures and/or tables, authored or reviewed drafts of the article, and approved the final draft.

## Human Ethics

The following information was supplied relating to ethical approvals (i.e., approving body and any reference numbers):

The study was approved by the University of Urbino Human Research Ethics Committee (approval reference number: VN21-10072019).

## Data Availability

The raw data is available in the Supplementary File.

## Supplemental Information

Supplemental information for this article can be found online at http://dx.doi.org/10.7717/peerj.17158#supplemental-information.

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
