# Peer review of "Are incremental exercise relationships between rating of perceived exertion and oxygen uptake or heart rate reserve valid during steady-state exercises?"

_PeerJ, doi:10.7717/peerj.17158_

## Round 0.1 · original submission · Major Revisions

While one reviewer expressed significant criticisms of the manuscript, particularly regarding the small sample size, I believe it is fair and appropriate to afford the authors an opportunity to address the concerns.

Reviewer 1 ·

Basic reporting

The paper could benefit from a good edit for grammar, sentence structure, etc.

Experimental design

This paper attempts to address an important issue regarding the use of RPE in the prescription of aerobic exercise. In this study, eight young adult men completed a maximal GXT and four steady-state running tests (60% and 80% of HHR for either 15 min or 45 min). The intent of this research was to determine if the relationship between RPE and %HRR and %VO2R during incremental exercise persist during aerobic exercise of different durations (15 and 45 min) and intensities (60% and 80% of HRR).

I have several concerns about this study, which are described below. Overall, I don’t feel the results of this study expand our current understanding of the use of RPE during exercise training and prescription, or the limitations of using RPE.

Only 8 male subjects participated in this study. I don’t see a power analysis indicating that this was a sufficient number of subjects to find significant results when comparing two different intensities of exercise and two different durations of exercise. I have little confidence that the findings of this research can be extrapolated to other individuals because of the small sample size and because there were no female participants in this study. Sufficient numbers of males and females should be included in studies such as these. In addition, data should be analyzed for sex differences.

During the practice trials, the participants ran at speeds that corresponded to either 60% or 80% of VO2R yet the four steady-state trials were run at 60% to 80% of HRR. The authors presume there is a 1:1 relationship between %HRR and %VO2R. This is a common assumption (partly because the ACSM promotes this) but a recent paper reported that the %HRR-%VO2R relationship did not coincide with the line of identity (intercept = 0, slope = 1) (https://doi.org/10.3390/ijerph192416914) which was also reported elsewhere (https://doi.org/10.1249/MSS.0000000000002434). Why not just have the participants run at 60% and 80% of HRR? How long were the practice trials?

During the 15 min and 45 min steady-state runs, the treadmill speed was continuously adjusted in order to maintain a predetermined HR (Lines 238-239). Thus, the authors forced the data by adjusting treadmill speed rather than select the treadmill speed and report HR, VO2, and RPE values over time (15 min or 45 min). Graphs of the VO2, HR, and RPE responses to the exercise tests over time (15 min and 45 min) would be valuable and the interpretation of the data would be more applicable to the purpose of the paper to investigate the relationships with different durations of exercise. There was no indication in the paper as to the adjustments in treadmill speed of the course of the 15 min and 45 min runs nor was there any report about how the adjustment is treadmill speed affected the VO2 response.

Participants reported RPE values one time at the end of the steady-state test (i.e., at the end of 15 min and the end of 45 min) (Lines 247-249). How did the authors control for participants being biased by the RPE values they self-reported on previous tests? Why were RPE values not self-reported more frequently (to coincide with HR and VO2 measurements) to determine trends in RPE over time? This would better reflect the relationship between RPE and HR and between RPE and VO2 over the different durations of the exercise test. A single RPE value is difficult to interpret – especially when trying to describe the relationship between RPE and HRR or VO2R.

Validity of the findings

See some of the comments in Section 2 about experimental design. Overall, results based on single data points over 15 and 45 min runs from 8 males subjects can hardly be extrapolated to other individuals.

Additional comments

Participants were “experienced” runners rather than “novice” runners. There is no discussion about the application of this data to the general population. Likewise there is no discussion about how the inclusion of only 8 male subjects affects the interpretation and application of this data.

Other than prior exercise and diet, there is little information about other factors that could affect HR responses and RPE values, such as room temperature and humidity, altitude, if a fan was place in front of the treadmill to help cool the participant during their 15 min and 45 min runs.

Participants were involved in this study for 7 days (Line 152). During the first three days, participants completed their maximal GXT and two practice trials. The four experimental trials were completed thereafter with at least 3 days separating each test. I don’t see how this adds up to 7 days.

Line 161. This sentence read as if the participants were asked “to take alcohol and/or caffeine the day before the test and on the testing days”. It should read, “Participants were asked to avoid changes in their eating habits, vigorous physical activity and consumption of alcohol and caffeine the day before and the day of testing.”

Maximal Exercise Test. The protocol for the maximal GXT appears very complicated. There are other standardized protocols that allow participant to reach VO2max within 10-12 minutes. My main concern on how the maximal GXT for each participant was designed, is that it appears that the duration of each stage would be different for each participant. Since RPE was recorded during each stage, each participant would have been jogging at a given speed for different amounts of time. In addition, the increment in speed was different for each participant. Duration of each stage and incremental changes in speed could affect the RPE reported by the participant. There is no indication in the paper of the duration of the stages or speed increments for the participants.

The exercise tests were repeated if the actual HR did not coincide with the target HR (Lines 242-247). Not forcing the HR response by keeping the running speed constant over the duration of the test (see above comment) would eliminate this problem. How many subjects repeated tests?

VO2 and HR data during the last 5 minutes of each steady-state run were averaged and “considered steady-state values.” I don’t see any definitions of “steady state” such as less than a certain about of change over a certain amount of time.

I don’t understand the purpose of “predicting” RPE values based on the RPE values reported during the maximal GXT (Lines 269-271) and then compared to the actual RPE values reported at the end of the 15 min and 45 min runs. Using RPE values during non-steady state incremental workloads to predict steady state RPE values during 15 and 45 min runs doesn’t make sense considering the various things that influence RPE. Simply reporting HR, VO2, and RPE values over the duration of the 15 min and 45 min test would have been more meaningful and applicable.

Reviewer 2 ·

Basic reporting

This study presents a well-written, novel, and intriguing exploration, aiming to assess whether RPE relationships with physiological parameters (i.e., %HRR and %VO2R) derived from incremental exercise can also be applied to prolonged exercise and to verify if the duration and intensity of prolonged exercise affect these relations.

Experimental design

The experimental protocol is well-designed. While the study holds promise, my primary concern centers around the relatively small sample size and its potential implications on the reliability and generalizability of the results, conclusions, and data extrapolation. Adequate consideration and discussion of the limitations imposed by the sample size are crucial to strengthen the study's overall impact and applicability.

Validity of the findings

While the presented results are notably intriguing, the discussion appears somewhat unsupported by the presented data. This discrepancy could be attributed to the limited power of results owing to the small participant size. It is recommended that the authors acknowledge and discuss the impact of the small sample size on the interpretability and generalizability of the findings. Providing such context will enhance the credibility and clarity of the study's contributions to the existing literature.

Additional comments

Specific Comments:
1. Sample Size: My primary concern revolves around the sample size. Have the authors conducted a calculation for the minimal sample size? The current sample size appears to be low, and the absence of an explanation exacerbates the issue. This aspect is critical as a small sample size tends to favor the null hypothesis. The authors must address this concern comprehensively to enhance the robustness of their results.
2. Sample Recruitment Process: It would be beneficial for readers to gain insight into the sample recruitment process. Was a convenience sample employed? Additionally, the rationale behind the exclusive selection of young male participants should be elucidated.
3. Factorial ANOVA: The inclusion of intensity in the factorial ANOVA seems questionable. This comparison needs more justification in the rationale of the study. I recommend opting for a two-way repeated measures ANOVA with modality and duration factors for each intensity. This approach would better address the existing gap in the literature, particularly considering the apparent influence of intensity on the dependent variable. Given the limited number of participants, such precision in the analysis is crucial for drawing meaningful conclusions.

Reviewer 3 ·

Basic reporting

The study investigates the relationship between subjective effort perception and oxygen consumption or heart rate reserve during constant load exercise. The research premise is intriguing and provides valuable insights to the existing literature. However, it seems to be based on secondary data from a study published in the same journal last year. Nevertheless, the analyses are well-executed, offering limited but valuable contributions to the literature, particularly in complementing the information from the previous study.
It is important to mention that the study uses clear and grammatically satisfactory language. The introduction provides essential context for the research's proposal and development. The references include up-to-date and classical references on the addressed topic. The manuscript structure adheres to the journal's standards. The absence of figures makes understanding the methodology slightly more challenging, a thoughtful inclusion of the study design spanning the 7-day testing period would undoubtedly enhance clarity. While no figures presenting the results were included, the tables suffice for visualizing the findings. In addition, the raw data is included as supplementary material, fostering transparency and offering readers an opportunity for deeper engagement with the study's outcomes.

Experimental design

These findings seemingly stem from supplementary data gleaned from a preceding study documented in PeerJ (Ferri Marini et al., 2022). Despite the formulation of a well-defined research question and the exploration of a pertinent subject, the methodology reveals certain limitations, notably a restricted number of volunteers and a homogenous participant profile, as underscored in the study's limitations section. This, in turn, curtails the generalizability of the results to broader populations. However, the conducted analyses were rigorous and well-described, with sufficient information for replication if complemented with details from the previous study by the same group (Ferri Marini et al., 2022).

Validity of the findings

The study's findings suggest that subjective effort perception does not demonstrate the same correlation with physiological parameters (HR reserve and VO2 reserve) observed in incremental testing, particularly when prolonged efforts are involved. This indicates the necessity of incorporating other physiological parameters associated with RPE for exercise prescription and monitoring. The significance of these results is particularly evident in methodological discussions and the critical examination of other studies, enriching the comprehension of outcomes influenced by the use of subjective effort perception, especially in extended aerobic exercise interventions. However, it is imperative to acknowledge that the results are highly constrained to a specific population of young, active, and healthy individuals and may not be entirely representative, given the limited number of participants. Nevertheless, the study provides pertinent information that contributes to future discussions and the design of subsequent studies.

---

## Round 0.2 · accepted · Accept

in line 503-504 (tracked-changes manuscript) please change "...even though not reaching a statistical significance due to small sample size,... " to "...even though not reaching statistical significance,... " - Please don't insinuate that the result would have been significant with a larger sample size. This is not known and pure speculation.

Reviewer 3 ·

Basic reporting

The modifications made were sufficient to prepare the manuscript for publication.

Experimental design

The writing modifications and additions were sufficient to meet the requirements for methodological quality.

Validity of the findings

The clarification regarding the protoco and the exposition of limitations were sufficient to support the conclusion.

Additional comments

I am pleased with the revisions made to the manuscript. It appears that all of the reviewers' suggestions were carefully considered and implemented to the best of their ability, resulting in a clearer and more comprehensible manuscript.